# Reduced Plasma Levels of Very-Long-Chain Dicarboxylic Acid 28:4 in Italian and Brazilian Colorectal Cancer Patient Cohorts

**DOI:** 10.3390/metabo8040091

**Published:** 2018-12-06

**Authors:** Paul L. Wood, Michelle M. Donohue, John E. Cebak, Taylor G. Beckmann, Márcia Cristina Fernandes Messias, Laura Credidio, Cláudio Saddy Rodrigues Coy, Patrícia de Oliveira Carvalho, Sara Crotti, Sara D’Aronco, Emanuele D.L. Urso, Marco Agostini

**Affiliations:** 1Metabolomics Unit, College of Veterinary Medicine, Lincoln Memorial University, 6965 Cumberland Gap Pkwy, Harrogate, TN 37752, USA; michelle.donohue@lmunet.edu (M.M.D.); john.cebak@lmunet.edu (J.E.C.); 2Department of Medicine, DeBusk College of Osteopathic Medicine, Lincoln Memorial University, 6965 Cumberland Gap Pkwy., Harrogate, TN 37752, USA; taylor.beckmann@lmunet.edu; 3Laboratory of Multidisciplinary Research, São Francisco University (USF), Bragança Paulista, São Paulo 12916-900, Brazil; marcia_cfmessias@hotmail.com (M.C.F.M.); patricia.carvalho@usf.edu.br (P.d.O.C.); 4Department of Surgery, University of Campinas (UNICAMP), Campinas, São Paulo 12916-900, Brazil; laurabio@gmail.com (L.C.); claudiocoy@gmail.com (C.S.R.C.); 5Nano-Inspired Biomedicine Lab, Istituto di Ricerca Pediatrica-Città della Speranza, 35127 Padua, Italy; s.crotti@irpcds.org (S.C.); s.daronco@ircpds.org (S.D’A.); m.agostini@unipd.it (M.A.); 6First Surgery Clinic, Department of Surgical, Oncological and Gastroenterological Sciences, University of Padua, Padua 35128, Italy; edl.urso@unipd.it

**Keywords:** colorectal cancer, inflammation, very-long-chain dicarboxylic acid 28:4, familial adenomatous polyposis, high-resolution mass spectrometry, cancer biomarker

## Abstract

Background: There are currently no blood-based biomarkers for early diagnosis of colorectal cancer. Previous research has suggested that very-long-chain dicarboxylic acid (VLCDCA) 28:4 might be such a biomarker. Methods: Using high-resolution mass spectrometry, we analyzed VLCDCA 28:4 in the plasma of colorectal cancer patients in Italian [*n* = 62] and Brazilian [*n* = 52] cohorts. Additionally, we investigated individuals diagnosed with familial adenomatous polyposis (FAP; *n* = 27), one of the most important clinical forms of inherited susceptibility to colorectal cancer. *Results:* Decrements in plasma levels of VLCDCA 28:4 were monitored in colorectal cancer patients. These decreases were independent of the stage of tumor development and the individual’s age. However, no decrements in VLCDCA 28:4 were monitored in FAP patients. Conclusions: The plasma levels of VLCDCA 28:4 represent a potential biomarker of sporadic colorectal cancer. In addition, it is possible that resupply of this anti-inflammatory lipid may represent a new therapeutic strategy for CRC and inflammatory disorders.

## 1. Introduction

Colorectal cancers (CRC) are the 3rd most prevalent cancer globally and represent multi-factorial disorders with sporadic cases (>90%) predominating [1]. The goal of healthcare systems is the early detection of CRC so that surgery can halt the progression to advanced CRC, which is much more difficult to treat [2]. The current gold standard for CRC detection is a colonoscopy every ten years for men and women starting at the age of 50, or a sigmoidoscopy at more frequent intervals for qualifying persons. Less expensive tests for routine annual screening include guaiac-based fecal occult blood tests and fecal immunochemical tests for hemoglobin [3], and next-generation sequencing [4]. However, these tests show low sensitivity and require further confirmations. As a consequence of these limitations, there are currently no blood-based biomarkers for the early detection of CRC.

A blood-based diagnostic CRC biomarker should possess sensitivity and specificity and should correlate with the severity of the disease. In this regard, plasma peptides [5], miRNAs [6], glycerophospholipids [7], and decanoic acid [8] are currently under investigation as potential blood biomarkers for the early diagnosis of CRC. In addition, a family of 28 to 36 carbon lipids, possessing anti-inflammatory and anti-proliferative properties in tissue culture [9], has been found to be decreased in the plasma of patients with CRC [10,11,12]. Of these lipids, decrements in the anion with *m/z* 445.3323, corresponding to the [M-H]^−^ anionic species of C_28_H_46_O_4_ with an exact mass of 446.3396, correlated best with a significantly increased risk for CRC [10,11,12]. While the structure of this lipid remained elusive, we recently identified the lipid as VLCDCA 28:4n6 [13]. This dicarboxylic acid was monitored in a number of human biofluids (plasma, synovial fluid, pleural fluid, cerebrospinal fluid, and aqueous humor), and decreased plasma levels were measured in 9 Russian and 26 American CRC patients [13].

To further monitor for ethnic differences in this potential CRC biomarker, we analysed plasma samples from Italian and Brazilian CRC patient cohorts. This also further augmented the number of patients evaluated, thereby allowing for the detection of potential biochemically distinguished subgroups within a given clinical diagnosis.

## 2. Results

### 2.1. VLCDCA 28:4 Plasma Levels in Italian and Brazilian CRC Cohorts

In both the Italian and Brazilian CRC patient cohorts, there was a statistically significant reduction in the levels of circulating VLCDCA 28:4 (Figure 1). This is clearly demonstrated by the box-and-whiskers plots where the 25th : 50th : 75th percentiles were 0.448 : 0.629 : 0.809 (Italian controls) compared to 0.166 : 0.310 : 0.540 (Italian CRC). Similarly, for the Brazilian cohorts the percentiles were 0.559 : 0.833 : 1.10 (controls) compared to 0.273 : 0.427 : 0.573 (CRC). The percentile plots of the plasma levels of VLCDCA 28:4 also clearly demonstrate the leftward shift in the CRC curves (Figure 1). There was a statistically significant difference (*p* < 0.01) between the control plasma levels of VLCDCA 28:4 for the 2 different ethnic cohorts. It remains to be established if this is a true ethnic difference, a dietary difference, and/or a methodology issue with measuring relative rather than absolute plasma levels. Once an absolute quantitation assay has been validated, this issue can be addressed. Of note, there was no statistically significant difference between the 2 CRC patient cohorts for plasma levels of VLCDCA 28:4.

The plots of plasma VLCDCA 28:4 levels relative to subject age are presented in Figure 2. These data clearly indicate that the plasma levels of VLCDCA 28:4 are independent of a subject’s age. Similarly, we did not detect any gender differences in the decrements of plasma VLCDCA 28:4, as has been previously reported [11,12].

### 2.2. Plasma VLCDCA 28:4 in Stage I–IV CRC Patients

The subdivision of the Italian CRC patient cohort into the different stages of tumor development demonstrated that VLCDCA 28:4 is decreased at all stages of tumor development (Stages I to IV) (Figure 3). These data are consistent with previous studies before the identity of C_28_H_46_O_4_ was obtained [13]. Previous studies [10] demonstrated that C_28_H_46_O_4_ was decreased at all stages of CRC diagnosis and that plasma levels remained decreased even after the surgical removal of the cancer. This led to the conclusion that C_28_H_46_O_4_ is not of tumor origin and represents a host anti-inflammatory lipid that may possess chemopreventive properties [9,13]. The host source of this dicarboxylic acid remains to be defined.

### 2.3. Plasma VLCDCA 28:4 in FAP Patients

The data obtained with different CRC patient populations demonstrated that VLCDCA 28:4 represents a potential blood biomarker for CRC. We also wanted to understand if this anti-inflammatory lipid is decreased in the plasma of patients with an established genetic predisposition for developing CRC. FAP patients start to develop hundreds to thousands of adenomatous polyps beginning in early adolescence and approximately 95% of patients develop CRC by age 50. The most relevant problem in FAP disease management is the difficulty in understanding the temporal window in which the switch between adenoma and malignant carcinoma occurs. To this end, we analyzed plasma samples of patients diagnosed with FAP and found that VLDCA 28:4 levels are not altered in this patient population (Figure 4).

## 3. Discussion

In previous studies of patients with CRC, the levels of C_28_H_46_O_4_ were not restored to normal following surgery, suggesting that they are not of tumor origin, or that tumor signaling suppresses VLCDCA 28:4 production [10]. These data, along with our findings, suggest that VLCDCA 28:4 is maintained at a homeostatic level in the bloodstream and that decrements in the levels of this anti-inflammatory lipid can result in the development of CRC. Inflammation plays an important role in the development of malignancies, such that Inflammatory Bowel Disease (IBD) [14] and Irritable Bowel Syndrome (IBS) [15], are associated with an increased risk of developing CRC, while aspirin [16] and non-aspirin NSAIDs [17] have been shown to decrease the risk of developing CRC. However, the CRC patient population is an heterogeneous group, since CRC can develop via different biochemical pathways. This is reflected in the data indicating that NSAIDs are chemopreventive for a subpopulation (18 to 44%) of individuals over the age of 40 years, at risk for CRC [17]. This heterogeneity is strongly influenced via a variety of high and low penetrance genes [18,19] and by a large number of environmental factors [20].

At the anatomical level, adenomas are extremely heterogeneous with regard to histological types (hyperplastic polyps and sessile serrate adenoma/polyps), as well as polyp numbers and sizes [21]. Furthermore, in patients with more than one primary tumor, the tumors express significantly different mutated genes [22].

In toto, this heterogeneity in both the genotype and phenotype of CRC patients strongly supports the concept that CRC develops via a diverse number of pathways and that no single biomarker is likely to detect all CRC cases. This is the case for VLCDCA 28:4, which detects a significant subset of sporadic CRC patients, but not all CRC patients. Our data also indicate that in FAP patients, the genetic mutations resulting in CRC do not involve VLCDCA 28:4.

Inflammation-cancer cross-talk involves multiple cellular and molecular pathways that possess both tumor-promoting and tumor-suppressing effects. The role(s) of VLCDCA 28:4 in resolving sustained inflammatory mechanisms require further elucidation; however, it is tempting to speculate that restoring VLCDCA 28:4 plasma levels in individuals with lowered levels may support an endogenous anti-inflammatory pathway. This is important since reduced levels of VLCDCA 28:4 have also been monitored in kidney and pancreatic cancers [13]. The restoration of VLCDCA 28:4 may also provide chemopreventive actions against CRC development and may potentially be antineoplastic. We are therefore currently designing prodrugs of VLCDCA 28:4 to evaluate these potentials.

In conclusion, our data and the previous evaluation of a large population of individuals undergoing colonoscopy [12] indicate that plasma VLCDCA 28:4 measurements have the potential to detect a significant subset of sporadic CRC patients. However, it is now critical to establish an absolute quantitation assay so that inter-laboratory assay differences will be minimized, and then to evaluate a much larger population of CRC patients and subjects at risk of developing CRC. These studies will be needed to establish a reliable cut-off between control and CRC VLCDCA 28:4 measurements. To achieve this goal, we have synthesized VLCDCA 28:4 and [^2^H_4_] VLCDCA 28:4, thereby providing the critical standards to establish and validate an absolute quantitation assay.

## 4. Materials and Methods

### 4.1. Clinical Samples

De-personalized plasma samples were obtained from controls, colorectal cancer patients, and familial adenomatous polyposis patients. Patient demographics are presented in Table 1. TNM classification of tumor staging was based on the criteria of AJCC/UICC staging system 7th edition [23]. Control samples were from volunteers and the CRC samples were obtained prior to chemotherapy.

This study was conducted according to the principles expressed in the Declaration of Helsinki. All blood samples were collected under the full ethical approval of the ethics committee and only after informed consent had been obtained from all of the patients enrolled in the study. Italian plasma samples were obtained from the Tissue Biobank of the First Surgical Clinic of Padua Hospital (Italy). The protocol was approved by the ethics committee of the institution (Comitato Etico del Centro Oncologico Regionale, Approved Protocol Number: P448). The ethics committee of the São Francisco University (CAAE: 51356315.5.0000.5514) and the Faculty of Medical Sciences of the State University of Campinas (CAAE: 08287012.0.0000.5404) approved the sample collection for the Brazilian Plasma samples.

### 4.2. VLCDCA 28:4 Analyses

The samples from both studies were analyzed at Lincoln Memorial University on the same instrument. To 100 microliters of plasma was added 1 mL of methanol containing 1 nanomole of the stable isotope internal standard, [^2^H_28_]dicarboxylic acid 16:0 and bromocriptine. Next, 1 mL of water and 2 mL of methyl-tert-butyl ether were added, and the tubes vigorously shaken at room temperature for 30 min. After centrifugation at 4000× *g* for 30 min at room temperature, the upper organic layer was isolated and dried by centrifugal vacuum evaporation at room temperature. The dried extract was dissolved in isopropanol : methanol : chloroform (4:2:1) containing 7 mM ammonium acetate for direct infusion electrospray mass spectrometric analysis [13,24].

Electrospray ionization (ESI) with a sheath gas of 12, a spray voltage of 3.7 kV, and a capillary temperature of 321 °C was used for the acquisition of high-resolution (<3 ppm mass error) negative ion ESI data, with an orbitrap mass spectrometer (Thermo Q Exactive). The anions for the stable isotope internal standard and endogenous VLCDCA 28:4 were acquired. The anion of bromocriptine was used to monitor for potential mass axis drift as a quality control. Between injections, the transfer line was washed with successive 500 µL washes of methanol and hexane/ethyl acetate/chloroform (3:2:1).

### 4.3. Statistical Analysis

The data are presented as R values, which are the ratios of the peak intensity for endogenous VLCDCA 28:4 to the peak intensity of the internal standard per 100 microliters of plasma. For multiple comparisons, ANOVA was followed by the Tukey–Kramer test for post-hoc analyses (Microsoft Excel). For simple comparisons, a Student’s t-test (Microsoft Excel, Seattle, WA, USA), assuming equal or unequal variances, was used to determine significant differences in levels of VLCDCA 28:4, subsequent to an F-test (Microsoft Excel) to determine if the variances between groups were statistically different.

## 5. Conclusions

VLCDCA 28:4 is a promising blood-based biomarker for CRC. The next major step will be to establish an absolute quantitation assay. This will provide more accurate results, allow for the determination of potential biochemically defined subgroups, and thereby provide reliable estimates of sensitivity and specificity parameters for this biomarker. In addition, based on the potent anti-inflammatory actions of this lipid [9,13], it will be important to evaluate the anti-inflammatory and anti-proliferative actions of VLCDCA 28:4 prodrugs.

## 6. Patents

Identification and use of very long chain dicarboxylic acids for disease diagnosis, chemoprevention, and treatment. (Paul L. Wood) USPTO 15/284,219.

## Figures and Tables

**Figure 1 metabolites-08-00091-f001:**
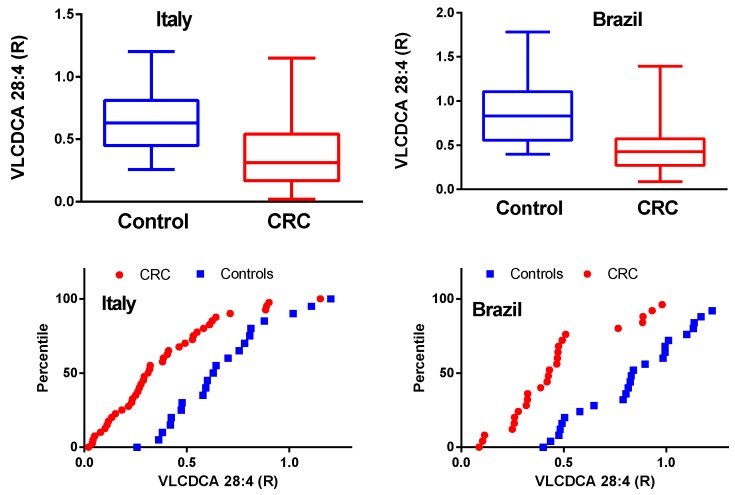
Relative plasma levels of very-long-chain dicarboxylic acid (VLCDCA) 28:4 in Italian and Brazilian colorectal cancers (CRC) cohorts, and age-matched controls, are presented in box-and-whiskers and percentile plots. For the box-and-whisker plots, the horizontals within the boxes are the medians (50th percentile) while the upper and lower lines of the boxes represent the 75th and 25th percentiles respectively. The whiskers represent the minimum and maximum levels. *p* = 0.00014 (Italy) and 0.000041 (Brazil).

**Figure 2 metabolites-08-00091-f002:**
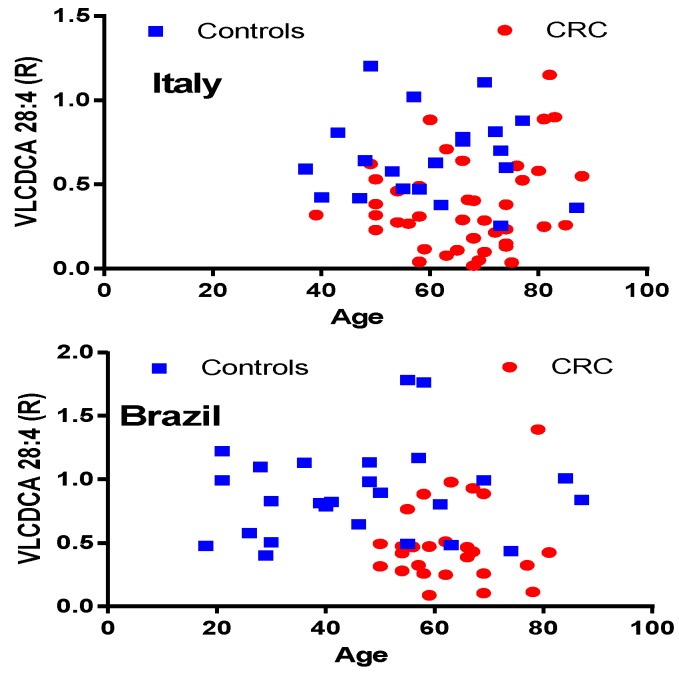
Presentation of the lack of age-dependent changes in plasma levels of very-long-chain dicarboxylic acid (VLCDCA) 28:4.

**Figure 3 metabolites-08-00091-f003:**
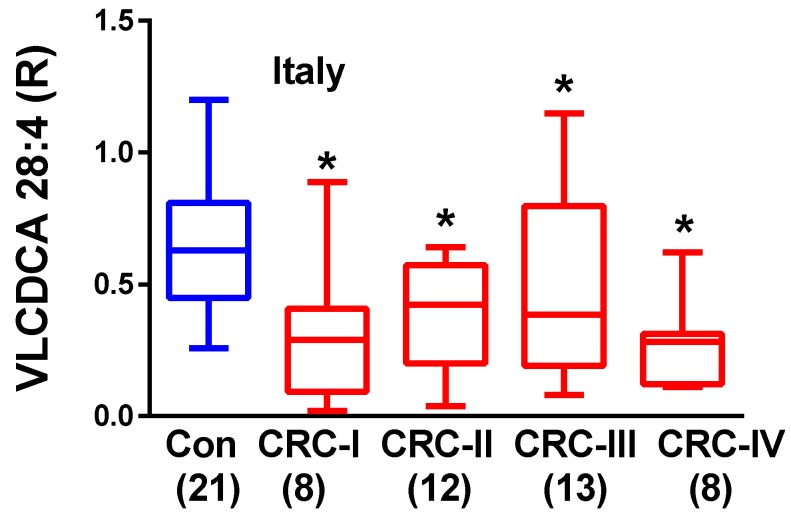
Relative plasma levels of very-long-chain dicarboxylic acid (VLCDCA) 28:4 in the Italian colorectal cancers (CRC) cohort and age-matched controls are presented in box-and-whiskers plots based on the stage of the CRC. For the box-and-whisker plots, the horizontals within the boxes are the medians (50th percentile) while the upper and lower lines of the boxes represent the 75th and 25th percentiles respectively. The whiskers represent the minimum and maximum levels. The brackets contain the N values per cohort. *, *p* < 0.01 vs. controls.

**Figure 4 metabolites-08-00091-f004:**
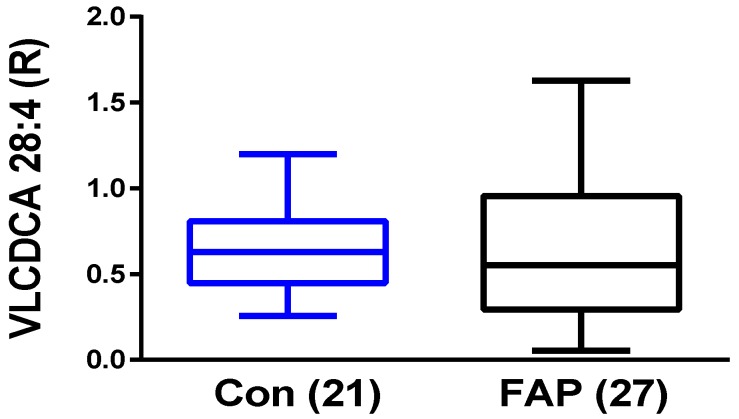
Relative plasma levels of very-long-chain dicarboxylic acid (VLCDCA) 28:4 in the Italian familial adenomatous polyposis (FAP) cohort and controls (Con) are presented in box-and-whiskers plots. The horizontals within the boxes are the medians (50th percentile) while the upper and lower lines of the boxes represent the 75th and 25th percentiles respectively. The whiskers represent the minimum and maximum levels. The brackets contain the N values per cohort.

**Table 1 metabolites-08-00091-t001:** Patient demographics for plasma very-long-chain dicarboxylic acid (VLCDCA) 28:4 measurements.

Group	Age (Yr. ± SD)	Age Range	N	Number of Females
Italy—Controls	60.3 ± 13.4	37–87	21	11
Italy—Colorectal cancers (CRC)	66.4 ± 11.6	39–88	41	23
Italy—Familial adenomatous polyposis (FAP)	45.0 ± 14.1	21–59	27	23
Brazil—Controls	46.7 ± 19.1	18–87	26	14
Brazil—CRC	63.0 ± 8.8	50–81	26	9

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
