# Peer review of "Reduced Plasma Levels of Very-Long-Chain Dicarboxylic Acid 28:4 in Italian and Brazilian Colorectal Cancer Patient Cohorts"

_metabolites, 2018, doi:10.3390/metabo8040091_

Reviewer 1 Report

Authors have developed such an interesting work regarding to the potential use of very-long-chain dicarbozylic acid 28:4 as a biomarker of colorectal cancer. However, the Discussion section can be considerably improved since it does not reflect the relevance of their findings:

1. Authors have analyzed plasma levels of VLCDCA 28:4 at all stages of CRC. Have they considered to analyze them on patients recovered from this disease in order to consider this biomarker as a potential biomarker of recurrency? This should be included on Conclusions as a likely future research to perform.

2. The implications of the results obtained from FAP patients have not been discussed. Why would be the most likely explanation?

3. In line with my previous statement, lines 188-189 should be rewritten, moved or removed. The use of VLCDCA 28:4 prodrugs have been not previously mentioned on the paper, and since authors considere that this is important it should be added to the Discussion section, not just mentioned on the Conclusions. However, it should be emphasized that this prodrugs would not be useful for every colon cancer patient, and consequently its therapeutic potential might be quite limited. This is the reason why I suggest removing this two lines.

4. References aported by authors could be improved with recent publications. Some suggestions are:

Mármol et al Colorectal carcinoma: a general overview and future perspectives in colorectal cancer. Int J Mol Sci 2017

Meester et al Optimizing colorectal cancer screening by race and sex: Microstimulation analysis II to inform the American Cancer Society colorectal cancer screening guideline. Cancer 2018

Issa et al Colorectal cancer screening: An updated review of the available options. World J Gastroenterol 2017.

Author Response

Added to results and  Discussion

FAP added to Discussion

Prodrug issue added to Discussion

More recent reference added.

Reviewer 2 Report

The manuscript by P. Wood et al. shows the potential application a very long chain dicarboxylic acid (VLCDCA) as a colorectal cancer (CRC) diagnostic biomarker. The authors investigated the relative levels of this compound by means of direct infusion electrospray ionization-mass spectrometry in cohorts comprised of CRC patients and controls from Brazil  (n= 52) and Italy (n=62). The relative levels of this compound were also evaluated in a small cohort of Italian individuals with familial adenomatous polyposis (n=27).The corresponding author has just published in the same journal a paper describing the use of this compound as a potential biomarker for different cancers. The previous publication shows that VLCDCA 28:4 levels decreases in patients with pancreatic, colorectal and kidney cancer. The information provided in this manuscript is complementary to the previous study, but is not conclusive regarding the specificity of VLCDCA towards CRC diagnosis, and the patient cohorts are not large enough to be considered a validation study.  In addition, some sections of this manuscript should be improved and there are results that have not been discussed.

Comments to the authors are divided based on the manuscript sections.

Introduction

Line 52-53: More information should be provided regarding the predomination of sporadic cases of CRC.

Line 59: “these tests shows” should be “these tests show”

Line 59-60: the sentence “As a consequence, there currently are…” should be revised.

Line 61: The statement is too simplistic. It is not clear if the authors want to address diagnostic or prognostic CRC biomarkers.  The statement could be improved based on the NIH, WHO and FDA definitions of biomarker. See for example: BiClinical Pharmacology and Therapeutics 69 (2001) 89–95.

Lines 62-64: It is suggested to discriminate the papers describing potential diagnostic biomarkers from those that suggest potential predictive biomarkers.

Line 66: The authors should provide the adduct ion that has m/z 446.3396.

Line 69: The biofluids listed by the authors are not ALL biological fluids, as indicated in the text

Results

Figure 2: The quality of the figure is should be improved.

Line 104: Reference 13 suggests that VLCDCA 28:4 decreases for other tumors. This would not make it an exclusive CRC potential biomarker. The authors should discuss this point.

Figure 3 suggests that there is no significant difference in the VLCDCA 28:4 R along disease progression by comparing the different CRC stages. These results are not addressed in the manuscript. They should be further discussed to improve the biological interpretation of the work.

Discussion

Lines 133-135: The authors suggest that changes in blood levels of markers like VLCDCA 28:4 will be helpful to monitor CRC onset and progression. However, Figure 3 does not show a trend in the lipid levels related to disease progression.

Lines 141-143: The results presented by the authors should be further discussed. The authors may provide quantitative information by indicating the number or percentage of false negatives obtained when comparing the lipid levels in CRC patients with the control median ratio.  Did the authors find any correlation between the lipid levels and gender?  The authors should also discuss if there is any correlation between CRC incidence and gender.

Methods

Were all samples from the different cohorts analyzed in the same center with the same instrument? This should be clarified.

At which temperature is performed the centrifugal vacuum evaporation?

Why the analytical method required to include bromocriptine to account for possible mass shift?

The authors describe the analytical method as a lipidomics study but they are just aiming at analyzing one lipid. This is a target study to determine relative levels of a specific compound.

Author Response

Line 52: predomination added

Line 59: fixed

Line 60: Fixed

Line 61 prognostic removed

Line 62: All diagnostic

Line 66: Adduct added

Line 69: All removed

Fig 2 enlarged

Line 104: Added to discussion

Fig 3:  Issue of CRC stages added to both the Results and Discussion sections

Line 133: Fixed in Results and Discussion

Line 141: Gender issue added to results

Methods: The 4 point were all corrected in the Methods

Reviewer 3 Report

The manuscript from Wood et al. displays the VLCDCA 28:4 level from two CRC cohorts of different ethnicities. In both groups the plasma concentration of this lipid drops in cancer patients in comparison to control group. Although, the data are not extensive the results are of interest as they contribute to a better understanding, if this VLCDCA is a useful biomarker for CRC regardless their ethnicity. As the authors stated, the measurement of total concentrations of this lipid instead of relative values, is essential for its use as a biomarker to define a cut-off between control and CRC.

Minor:

In the material and method part it has to be mentioned if the control group are volunteers or people with other conditions. CRC patients (especially UICC III and IV) are often treated with chemotherapy, please state, when the blood samples are taken, especially if the samples are taken before treatment or during treatment.

The patients quantity investigated in this study should already be mentioned in the abstract.                              

The discussion should be prolonged regarding:

Is this lipid also measured in relationship to other diseases, especially in IBD? 

There are several publications cited in the introduction that show data of VLC lipids in plasma from CRC patients, the results from this study have to be discussed in relation to the data from previous studies. How they fit, how they differ….

The data for FAP patients are not discussed at all, this has also to be included in the discussion part especially in relation to other literature.

Author Response

The steps required to define the cut-off have been added to the Discussion

Source of controls added to the methods

Patient numbers added to the abstract

The issues raised for the Discussion have all been added to the revised Discussion.

Round  2

Reviewer 1 Report

I consider that authors have improved the overall quality of the manuscript and therefore it can be published in the journal.

Author Response

The reviewer is satisfied.

Reviewer 2 Report

The manuscript by P. Wood et al. has significantly improved after revision. There are only minor points to be corrected.

Line 61: “there currently are” should be “there are currently”

Line 68: “the anion (445.3323) for the exact mass 446.3396 (C28H46O4,) correlated best” should be “the ion with m/z 445.3323 corresponding to the [M-H]- anionic species of C28H46O4 with exact mass 446.3396, correlated best…”

Line 141: “a heterogeneous” should be “an heterogeneous”

There is some contradictory information between lines 181 and 230: In line 181 the authors state that they have synthesized VLCDCA 28:4 and [2H4] VLCDCA 28:4, but in line 230 the authors state that the next steps are to synthesize a stable isotope analog.

Author Response

All minor points have been addressed and are tracked.